# Health problems and violence experiences of nurses working in acute care hospitals, long-term care facilities, and home-based long-term care in Germany: A systematic review

Andrea Schaller[1]*, Teresa Klas[1], Madeleine Gernert[1], Kathrin Steinbeißer[2,3]

1 Working Group Physical Activity-Related Prevention Research, Institute of Movement Therapy and Movement-Oriented Prevention and Rehabilitation, German Sport University, Cologne, Germany, 2 Faculty for Applied Healthcare Sciences, Technical University of Deggendorf, Deggendorf, Germany, 3 Institute of Health Economics and Health Care Management, Helmholtz Zentrum München, German Research Center for Environmental Health, Munich, Germany

* a.schaller@dshs-koeln.de

## Abstract

### Background

Working in the nursing sector is accompanied by great physical and mental health burdens. Consequently, it is necessary to develop target-oriented, sustainable profession-specific support and health promotion measures for nurses.

### Objectives

The present review aims to give an overview of existing major health problems and violence experiences of nurses in different settings (acute care hospitals, long-term care facilities, and home-based long-term care) in Germany.

### Methods

A systematic literature search was conducted in PubMed and PubPsych and completed by a manual search upon included studies' references and health insurance reports. Articles were included if they had been published after 2010 and provided data on health problems or violence experiences of nurses in at least one care setting.

### Results

A total of 29 studies providing data on nurses health problems and/or violence experience were included. Of these, five studies allowed for direct comparison of nurses in the settings. In addition, 14 studies provided data on nursing working in acute care hospitals, ten on nurses working in long-term care facilities, and four studies on home-based long-term care. The studies either conducted a setting-specific approach or provided subgroup data from setting-unspecific studies. The remaining studies did not allow setting-related differentiation of the results. The available results indicate that mental health problems are the highest for

**Data Availability Statement:** All relevant data are within the paper and its Supporting Information files.

**Funding:** AS received funding from the German Federal Ministry of Health, grant number 2520ZPK744. The funder had no role in study design, data collection and analysis, decision to publish, or preparation of the manuscript.

**Competing interests:** The authors have declared that no competing interests exist.

nurses in acute care hospitals. Regarding violence experience, nurses working in long-term care facilities appear to be most frequently affected.

## Conclusion

The state of research on setting-specific differences of nurses' health problems and violence experiences is insufficient. Setting-specific data are neccessary to develop target-group specific and feasible interventions to support the nurses' health and prevention of violence, as well as dealing with violence experiences of nurses.

## Introduction

Despite the international differences in health care systems, professional nursing is generally described as a crucial part of the health care system. It encompasses health promotion, disease prevention, and care of individuals of all ages with physical or mental illness, or with disabilities [1]. In the German care system, nursing care takes place in different settings, such as in hospitals, nursing homes, at home or in community-based institutions. Crucial settings for adult-based nursing care are acute care hospitals, long-term care facilities (LTC), and the patients' home.

In 2019, more than 19 million medical cases were treated in German acute medical care hospitals and rehabilitation facilities [2]. Beyond, more than 820,000 patients received long-term care (LTC) in LTC facilities. Around 980,000 patients received LTC by professional nurses in home-based settings [3]. With regard to the number of employees, there are currently more than 345,000 nurses working in acute care hospitals [4] and more than 600,000 nurses working in LTC facilities or in home-based LTC [5]. This means that around one million people are currently employed as professional nurses. Demand for this profession steadily rises due to demographic change and the increase of non-communicable diseases [6, 7].

However, about 75% of nurses assume not to be able to work in this profession until retirement under the given conditions [8]. Almost every second nurse thinks about leaving the profession several times a year [8–10]. One reason is that the profession appears to be associated with major health problems [11, 12], which is associated with a comparatively high number of sick days. Nurses in LTC facilities or home-based settings (24.1 days/year), or in hospitals (19.3 days/year) have considerably more sick days than employees in other occupational fields (16.1 days/year) [13]. In addition to health-related burdens, violence experiences are also considered to be a crucial occupational stress factor for nurses. This is reflected in the number of incidents as well as the the severity of the impact [14]. Violence experiences can include physical or verbal violence experiences, patient- or relatives related aggressions, and sexual harassment [15, 16]. Violence experiences can lead not only to physical harm, but also to mental health problems including impaired well-being or even symptoms of post-traumatic stress disorder [17–19]. Additionally, nurses' feelings of anxiety and anger due to violence experiences also go in line with less job satisfaction [20] and enhanced withdrawal intentions [21]. Studies show that at least 14% of nurses have been victims of violence in the past three months [22].

Despite the importance of nursing as an occupational field, little is known to date about specific health problems of nurses working in acute care hospitals, LTC facilities, or in home-based LTC. Tasks of nurses in these three areas differ considerably. Whereas tasks in the acute care hospital mainly focus on accompanying and supporting patients with acute medical treatment and recovery, the tasks in LTC for the elderly (e.g., in LTC facilities or home-based LTC)

are based on the concept of need for LTC according to the Social Code XI (German: "Sozialge-setzbuch XI") [23]. According to this, tasks mainly support the following sectors: support of mobility (e.g., changing positions in bed, transferring, moving around within the living area), support of cognitive and communicative abilities (e.g., orientation in time and place), reduce behavioral and psychological problems (e.g., self-damaging behavior, aggressive behavior toward other persons, depressive moods), promote activities of daily living (e.g., washing, (un) dressing, using a toilet), coping with illness- or therapy-related requirements and stress (e.g., with regard to medication, wound care), and fostering social contacts [23, 24]. Some tasks occur more frequently in the LTC facilities settings, while others may be more common in the home-based LTC setting. As nurses' daily working life differs over the settings, it is assumed that health problems and violence experiences might be different, too [25, 26]. This knowledge is an important requirement to develop target group-specific occupational health promotion and support programs for nurses. The research question for this review was: What are the health problems and violence experiences of nurses in acute care hospitals, LTC facilities, and home-based LTC?

## Methods

This systematic review was conducted following the international guidelines established by PRISMA (Preferred reporting items for systematic reviews and meta-analysis protocols) [27]. To ensure transparency and reproducibility, the systematic review protocol was registered in the International prospective register of systematic reviews (PROSPERO) (registration number: CRD42021231891).

### Search strategy

All potential articles from PubMed and PubPsych were obtained by electronic search. The search for both databases was performed on January 11th, 2021. Search terms used for relevant studies were built of the following keywords and Boolean operators (in cursive): (nurs* *OR* "professional care" *OR* "professional caregiver") *AND* (health *OR* violence) *AND* ("cross sectional" *OR* survey) *AND* (german*). Original studies in German or English language published between January 01st, 2010 and January 11th, 2021 were taken into account. Results were completed by a manual search upon included studies' references and health insurance reports.

### Inclusion and exclusion criteria

Articles that met the following inclusion criteria were considered for further analysis of (1) cross-sectional data, (2) target group or subgroup analysis: professional nurses in Germany, (3) setting: acute care hospital, LTC facilities and/or home-based LTC, (4) data on physical health, mental health and/or violence experience (physical or verbal violence experiences, patient-related aggressions, sexual harassment). Studies which met at least one of the following criteria were excluded: (1) longitudinal studies or validation studies, (2) qualitative studies, (3) studies outside of Germany. Additionally, we excluded studies addressing health issues of apprentices, supervisors, or managers.

### Quality assessment

To evaluate the selected articles and to identify the risk of bias of the included studies, the Joanna Briggs Institute's checklist for prevalence studies was applied [28]. This checklist includes nine items, which are answered with "yes", "no", "unclear", and "not applicable" respectively. The rating was conducted independently by two authors (MG, TK).

Disagreements in the ratings of the nine items were resolved after reconsideration and, if necessary, discussed with a third author (AS). For each study, the percentage of checklist items answered with "yes" was calculated. Studies were considered "low risk of bias" if the study scored ≥50% by fulfilling at least five quality requirements.

## Study selection, data extraction and synthesis

After eliminating duplicates, two authors (MG, TK) independently performed the title and abstract screening by using the software tool for systematic reviews "Rayyan" [29]. Subsequently, full-texts of the included studies were again independently assessed for eligibility and reasoned exclusions were recorded. Any disagreements were resolved by discussion and consensus with a third researcher (AS). The selection process was displayed in a PRISMA Flow Chart [27]. Data of the studies were separately extracted by two authors (MG, TK) and cross-checked in each case.

Extracted data included the setting in which the study was conducted (acute care hospital, LTC facilities, home-based LTC, cross-setting), author and publication year, sample size, sample characteristics (age, gender), the health problem and/or violence experience assessed in the study (physical health, mental health, and/or violence experiences) and the findings of the study related to the respective health problem. In the present study, professional nurses were considered to be qualified by graduation from an accredited school of nursing and by passage of a national licensing examination to practice nursing. Our definition of violence is based on the WHO definition of violence against patients or residents. This definition is acknowledged and accepted in the field of nursing and comprises emotional, physical and sexual forms of violence which cause harm or distress to the affected person [30, 31]. Some of the included studies did not contain all the aforementioned variables. In these cases, the available data were reported. Missing data was indicated by the note "not reported".

The extracted data were presented in four setting-related tables (acute care hospital, LTC facilities, home-based LTC, cross-setting). In this study, "acute care hospital" was considered as a setting where a patient needs immediate treatment and care (e.g., after an accident). Its goal lies on supporting patients with acute medical treatment and recovery. "LTC facilities" in this study were considered as nursing homes or professional nursing facilities. "Home-based LTC" was defined as the provision of nursing and domestic care of older people in need of LTC in their own homes. Both provision of LTC in "LTC facilities" and "home-based LTC" base on support with daily activities for people who experience a decline in self-care on a long-term basis [32]. The tables were used as a basis for a narrative synthesis of the key findings of the included studies.

## Results

### Selected studies

The initial search yielded 447 articles with 417 remaining after duplicates were removed. After screening titles, abstracts, and full texts, 17 studies were included. Additionally, six studies were identified by cross-reference and another six studies by health insurance reports. This resulted in a total of 29 articles (see Fig 1). Of these, 15 studies were found in PubMed and 8 studies in PubPsych, whereby six duplicates occured.

Of the 29 studies, ten studies addressed a setting-unspecific or cross-setting approach (Table 1). This implies that the sample could not or could only partially be assigned to the three settings studied. From these ten studies, it was possible to find specific subgroup data for settings in two studies, so that they were also assigned to the respective setting [33, 34]. Fourteen studies provided data on health and/or violence experiences of nurses working in acute

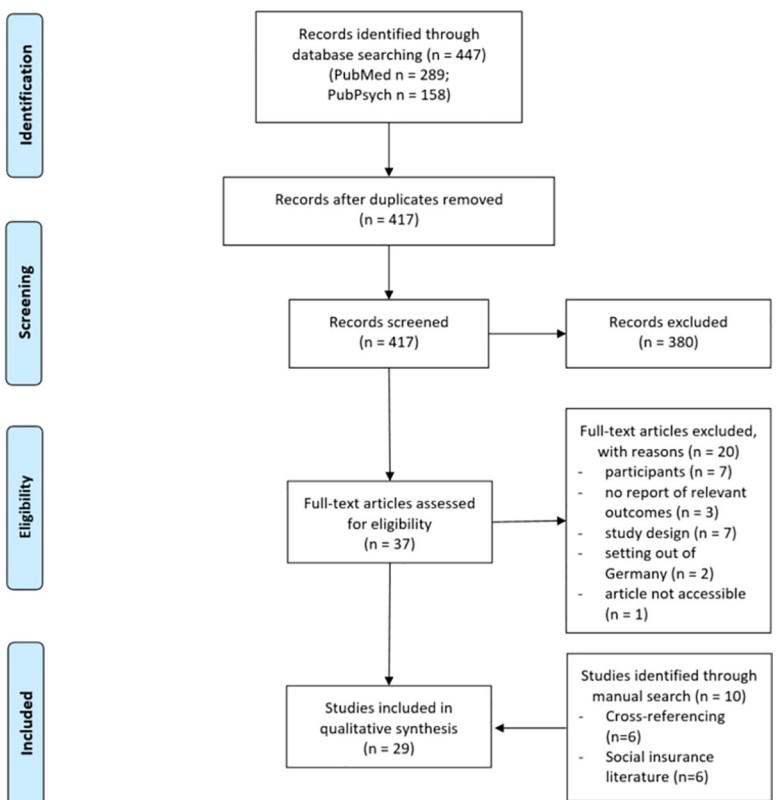

**Fig 1. PRISMA flow chart of the systematic literature search.**

care hospitals (Table 2), ten studies of nurses working in LTC facilities (Table 3), and four provided data regarding nurses working in home-based LTC (Table 4). The sample size varied between 20 nurses (LTC facilities) [35] to 355,988 (acute care hospitals) [36]. Overall, the proportion of female nurses in the studies ranged from 69.8% [37] to 93% [38] and the average age ranged between 26.5 years [37] to 45.7±11.4 [39]. The JBI Critical Appraisal Checklist [28] varied between 33% [15] and 100% [40, 41], with an average of 71%.

Regarding the health problems assessed, 23 of the 29 studies assessed mental health, twelve physical health, and nine violence experiences (multiple outcomes possible). Thereby, mental health was most frequently ascertained in the form of the latent construct burnout. Regarding physical health, the most frequently assessed aspects were musculoskeletal complaints [34, 40, 42]. Violence experiences were asked in four studies in terms of both physical and verbal violence [15, 16, 37, 43], and in terms of patient-related aggression [33], general experience of violence [22], or sexual harassment [44].

From a setting-specific perspective, the most frequently studied health problem in the acute care hospital setting was mental health (13 studies) [34, 35, 37, 38, 41, 45–51], whereas physical health was assessed in five studies, and violence problems in four studies. Seven of the ten studies with data on TLC facilities reported on mental or physical health, and three studies on violence experiences. Regarding home-based LTC, three subgroup data available assessed violence experiences, four studies mental, and three studies physical health. Mental health topics were also the most frequently addressed in the cross-setting and non-setting-specific studies.

**Table 1. Setting-unspecific/cross-setting—Summary of the studies included in the review.**

| Setting-unspecific/cross-setting | | | | |
|---|---|---|---|---|
| Author (year) | Sample | | Health problem, violence experience and related and outcome | Result |
| | Sample size (subgroups) | 1. Age [years].<br>2. Gender (female) | | |
| Diehl et al. (2020) [53] | 1316 (palliative care) | 1.<39: 26.5%; 40–49: 28.4%; >50: 45.1% | Physical health: subjective general health status | Physical health<br>Self-rated health [min: 0; max: 100]: M±SD = 72.86±16.94 |
| | | 2. 87.3% | Mental health: burnout | Mental health:<br>Burnout [min: 0; max: 100]: M±SD = 41.43±17.61 |
| Ehegartner et al. (2020) [40] | 1381 (27.8% hospital, 41.9% LTC facilities, 30.2% home-based long-term care) | 1. M±SD = 40.1 ±12.0 | Physical health: physician-diagnosed disease | Prevalence of physician-diagnosed disease (top 3)<br>Musculoskeletal diseases: 79.7%<br>Cardiovascular diseases: 38.8%<br>Mental impairments: 32.3% |
| | | 2. 81% | Mental health: physician-diagnosed disease | |
| Gencer et al. (2019) [33] | 167 (65.4% LTC facilities palliative care, 34.6% home-based palliative care) | 1. Median = 48 (Range = 23–62) | Physical health: subjective general health status | Physical health:<br>Prevalence of good/very good general health status: 64.2%<br>Mental health:<br>Prevalence of noticeably high strain: 27.6%<br>Prevalence of serious high strain: 27.6%<br>Violence:<br>Subgroup specific results |
| | | 2. 89.9% | Mental health: score | |
| Drupp & Meyer (2019) [36] | 355,988 (71.7% LTC; 24.9% nurses) | 1. Mean = 40.6 | Physical and mental health: physician-diagnosed disease | Physical health:<br>Respiratory diseases: 53.9 cases of work incapacity per 100 insured years<br>Musculoskeletal diseases: 39.5 cases of work incapacity per 100 insured years<br>Mental health:<br>Psychological diseases: 19.4 Cases of work incapacity per 100 insured years |
| | | 2. 85.5% | | |
| Lohmann-Haislah et al. (2019) [34] | 318 (Setting-unspecific subgroup of nurses in a study with several professions) | 1. 15–34: 21.4%; 34–54: 52.8%; >55: 25.9% | Physical health: Musculoskeletal problems, other health problems | Physical health:<br>Prevalence of musculoskeletal health problems (top 3):<br>Low back pain: 70%<br>Neck-shoulder pain: 64.3%<br>Pain in arms: 35.7%<br>Prevalence of other physical health problems (top 3):<br>Headache: 40.0%<br>Running nose/sneezing: 20.6%<br>Stomach and digestive problems: 18.0%<br>Mental health:<br>Prevalence of psychosomatic health problems (top 3):<br>General fatigue, tiredness, exhaustion: 57.8%<br>Physical exhaustion: 54.5%<br>Nervousness/irritability: 37.7% |
| | | 2. 90.4% | Mental health: psychosomatic complaints | |
| Schablon et al. (2012) [16] | 1178 (Setting-unspecific subgroup of nurses in a study with several professions, 23.8% head nurses, 76.2% nurses) | not reported for the subgroup of nurses | Violence: verbal, physical | Violence:<br>Prevalence of verbal violence in the past 12 months: 84%<br>Prevalence of physical violence in the past 12 months:: 61% |

(*Continued*)

**Table 1.** (Continued)

| Setting-unspecific/cross-setting | | | | |
|---|---|---|---|---|
| Author (year) | Sample | | Health problem, violence experience and related and outcome | Result |
| | Sample size (subgroups) | 1. Age [years]. 2. Gender (female) | | |
| Schablon et al. (2018) [43] | 884 (setting-unspecific subgroup of nurses in a study with several professions, 23.2% nurses with managerial role, 76.8% nurses without managerial role) | not reported for the subgroup of nurses | Violence: verbal, physical | Violence: Prevalence of verbal violence in the past 12 months: 96.6% Prevalence of physical violence in the past 12 months: 76.5% |
| Schmidt & Diestel (2014) [54] | 195 (cross-setting study including: nurses in a hospital and three nursing homes for the elderly) | 1. M±SD = 37.29 ±10.6 2. 85% | Mental health: burnout (emotional exhaustion, depersonalisation), depressive symptoms | Mental health: Emotional exhaustion [min: 1; max: 6]: M±SD = 2.2±0.79 Depersonalisation [min: 1; max: 6]: M±SD = 1.92±0.77 Depressive symptoms [min: 1; max: 5]: M±SD = 1.02±0.76 |
| Skoda et al. (2020) [55] | 1511 (Setting-unspecific subgroup of nurses in a study with several professions) | 1. not reported by the authors 2. 86.83% | Mental health: anxiety | Mental health: Generalized anxiety disorder: 11.41% |
| Weidner et al. (2017) [22] | 402 (Setting-unspecific subgroup of nurses in a study with several professions) | 1. not reported by the author 2. not reported | Violence | Violence: general experience of violence (5-fold likert scale: [min: 1; max: 5] 1: not at all; 5: very often): very often/often: 13.7% |

Five studies pursued a setting-comparative approach, of which data of two studies compared all three settings [35]. The two publications of Vaupel et al. [44, 51] enable a comparison between nurses working in acute care hospitals, LTC facilities, and home-based LTC, because they are based on the same primary data and structured analogously to each other. Both, the results of Otto et al. [35] as well as Vaupel et al. [44, 51] indicate that there are rarely differences in regard to physical health or well-being. However, nurses working in hospitals showed higher stress levels [35] and emotional exhaustion [35]. In contrast, exposure to violence experience was higher in nurses working in LTC facilities, followed by those working in home-based LTC [44, 51].

Two further studies compared health problems and/or violence experience of nurses working in LTC facilities or in home-based LTC [33, 35]. These data confirm that nurses working in LTC facilities seem to be more affected to violence experiences than nurses in home-based LTC. In contrast, nurses working in home-based LTC seem to be more frequently affected by burnout and cognitive stress symptoms than nurses working in LTC facilities [39, 50]. Comparing nurses working in acute care hospitals and in LTC facilities, findings indicate more physical and mental health complaints among nurses working in LTC facilities [50, 52].

## Discussion

To our knowledge, this is the first review that focused on summarizing and comparing major health problems and violence experiences of nurses working in acute care hospitals, LTC facilities, and home-based LTC in Germany. We must state that there are currently hardly any cross-setting primary studies providing a well-founded empirical data basis to compare major health problems and violence experiences. The available results indicate that mental health problems might be highest for nurses in acute care hospitals, whereas no setting-specific differences were identified with regard to physical health problems. Comparing nurses working in

**Table 2. Hospital—Summary of the studies included in the review.**

| Hospital | | | | |
|---|---|---|---|---|
| Author (year) | Sample | | Health problem, violence experience and related and outcome | Result |
| | Sample size (respondents) | 1. Age [years] | | |
| | | 2. Gender (female) | | |
| Aiken et al. (2012) [45] | 1508 (Subgroup of an international study) | not reported for the subsample nurses in hospitals | Mental health: burnout | Mental health:<br>Prevalence of burnout: 30% |
| Fischer et al. (2020) [46] | 576 | 1. <30: 28.6%; 31–40: 18.1%; 41–50: 26.4%; >51: 26.9% | Mental health: burnout | Mental health:<br>Prevalence of burnout symptoms (moderate to high): 50.4% |
| | | 2. 74.5% | | |
| Grobe & Steinmann (2019) [50] | 275,375 (Subgroup of a cross-setting study) | 1. not reported for the subsample nurses in hospitals | Physical and mental health: physician-diagnosed disease | Physician-diagnosed disease (top 3):<br>Muscular and skeletal diseases: 446 diagnoses per 100 insured years<br>Mental disorders: 428 diagnoses per 100 insured years<br>Respiratory deseases: 318 diagnoses per 100 insured years |
| | | 2. 80% | | |
| Kowalski et al. (2010) [41] | 959 | 1. M±SD = 38.0±9.8 | Mental health: burnout | Mental health:<br>Prevalence of burnout symptoms (moderate to high): 60% |
| | | 2. 87.9% | | |
| Lehmann-Willenbrock et al. (2012) [38] | 138 | 1. M±SD = 39.85±9.74 | Mental health: stress | Mental health:<br>Stress [min: 1; max: 6]: M±SD = 2.72±1.07 |
| | | 2. 93% | | |
| Lindner et al. (2015) [15] | 142 | not reported by the authors | Violence: verbal, physical | Violence:<br>Prevalence of verbal aggression in the past six months: 93%<br>Prevalence of physical aggression in the past six months: 46%<br>Prevalence of injuries due to aggression in the past six months: 34% |
| Lohmann-Haislah et al. (2019) [34] | 685 (Subgroup of a cross-setting study) | 1. 15–34: 17.5%; 34–54: 63.2%; >55: 19.2% | Physical health: Musculoskeletal problems, other health problems | Prevalence of musculoskeletal health problems (top 3)<br>Neck-shoulder pain: 65.3%<br>Low back pain: 63.8%<br>Pain in legs/feet: 34.6%<br>Prevalence of other physical health problems (top 3)<br>Headache: 43.0%<br>Running nose/sneezing: 27.4%<br>Stomach and digestive problems: 27.1%<br>Prevalence of psychosomatic health problems (top 3)<br>General fatigue, tiredness, exhaustion: 61.4%<br>Physical exhaustion: 53.5%<br>Sleep disorders: 52.3% |
| | | 2. 83.5% | Mental health: psychosomatic complaints | |
| Otto et al. (2019) [35] | 44 (Subgroup of a cross-setting study) | 1. M±SD = 29.45 ±11.16 | Physical health: score incuding physical functioning, role-physical, bodily pain and general health | Physical health:<br>Physical health score [min: 0; max: 100]: M±SD = 53.31±7.07<br>Mental health:<br>Mental health score [min: 0; max: 100]: M±SD = 43.72±9.84<br>Stress [min: 0; max: 48]: M±SD = 22.61±10.08 |
| | | 2. not reported by the authors | Mental health: score including vitality, social functioning, role-emotional; Stress | |
| Paffenholz et al. (2020) [47] | 834 | not reported by the authors | Mental health: concern for own health | Concern for own health in the context of the COVID-19 pandemic (5-fold likert scale: [min: 1; max: 5] 1: not at all; 5: very strongly)<br>Strongly: 21.8%<br>Very strongly: 9.5% |

*(Continued)*

**Table 2.** (Continued)

| Hospital | | | | |
|---|---|---|---|---|
| **Author (year)** | **Sample** | | **Health problem, violence experience and related and outcome** | **Result** |
| | **Sample size (respondents)** | **1. Age [years]** | | |
| | | **2. Gender (female)** | | |
| Raspe et al. (2020) [37] | 205 | 1. M±SD = 26.5±3.1 | Physical health: subjective general health status | Physical health: <br>Subjective general health status [min: 0; max: 100]: M±SD = 56.2±16.9 <br>Mental health: <br>Burnout [min: 0; max: 100]: M±SD = 57.1 ±16.3 <br>Violence <br>Prevalence of verbal aggression (at least 4x/year): 84% <br>Prevalence of physical aggression (at least 4x/year): 74% |
| | | 2. 69.8% | Mental health: burnout | |
| | | | Violence: verbal, physical | |
| Rothgang et al. (2020) [52] | 1,896 nurses in hospitals (subgroup of nurses in a study with several professions) | 1. Not reported by the authors | Physical health: complaints during/after work, physician-diagnosed disease | Complaints during/after work (top 3): <br>Prevalence of pain in arms/hands: 65% <br>Prevalence of low back pain: 64% <br>Prevalence of physical excaustion: 63% <br>Physician-diagnosed disease (top 3): <br>Muscular and skeletal diseases: 418 diagnoses per 100 insured years <br>Endocrine, nutritional and metabolic diseases: 248 diagnoses per 100 insured years <br>Diseases of the genitourinary system: 233 diagnoses per 100 insured years |
| | | 2. 0.7% (nurses working in LTC); | | |
| Vaupel et al. (2020) [51] | 123 | not reported for the subgroup of nurses in hospitals | Violence: verbal and nonverbal sexual harassment and violence | Violence: <br>Prevalence of nonverbal sexual harassment (at least one time/year): 50.0% <br>Prevalence of verbal sexual harassment and violence (at least one time/year): 76.0% <br>Prevalence of physical sexual harassment and violence (at least one tima/year): 47.0% <br>Mental health: <br>Prevalence of emotional exhaustion (at least one time/month): 69.0% <br>Prevalence of depressiveness (often/very often): 1.6% <br>Prevalence of psychosomatic complaints (every few months to daily): 97.5% <br>Prevalence of well-being (never to rarely): 13.8% |
| | | | Mental health: Burnout (emotional exhaustion), depressiveness, psychosomatic complaints, well-being (WHO 5) | |
| Wagner et al. (2019) [48] | 567 (Subgroup of a study with several professions) | not reported for the subgroup of nurses | Mental health: burnout | Mental health: <br>Burnout [min: 0; max: 100]: M±SD = 36.5 ±17.6 |
| Weigl & Schneider (2017) [49] | 13 (Subgroup of a study with several professions) | not reported for the subgroup of nurses | Mental health: burnout (emotional exhaustion, irritation) | Mental health: <br>Proportion of nurses reporting irritation: 69.2% <br>Proportion of nurses reporting emotional exhaustion: 53.8% |

LTC facilities with those in home-based LTC, mental and physical health problems in those working in LTC facilities appear to be higher. With regard to experiences of violence, nurses working in LTC facilities appear to be more frequently affected compared to those working in acute hospitals and home-based LTC.

Both the lack of comparative studies and the lack of setting-based studies, especially in home-based LTC, is surprising given the known remarkably differences in the working

**Table 3. Long-term care (LTC) facilities—Summary of the studies included in the review.**

| Long-term care (LTC) facilities | | | | |
|---|---|---|---|---|
| Author (year) | Sample | | Health problem, violence experience and related and outcome | Result |
| | Sample size (respondents) | 1. Age [years] 2. Gender (female) | | |
| Frey et al. (2018) [42] | 155 | 1. M±SD = 41±13 2. 83.3% | Physical health: low back pain, subjective general health status | Physical health: Lifetime prevalence of chronic back pain: 45.8% (women: 50.8%, men: 20.0%) Health status bad/ moderate: 38.1% |
| Gencer et al. (2019) [33] | 106 (Subgroup of a cross-setting study) | not reported for the subgroup of nurses in LTC facilities | Violence: patient aggression | Violence Patient aggression (6-fold likert scale: [min: 0; max: 6] 0: no strain; 5: high strain): M = 2.6 |
| | | | Physical health: subjective general health status | Physical health No results for subgroup reported |
| | | | Mental health: score | Mental health No results for subgroup reported |
| Grobe & Steinmann (2019) [50] | 52.016 (Subgroup of a cross-setting study) | 1. not reported for the subsample nurses in LTC facilities 2. 80% | Physical and mental health: physician-diagnosed disease | Physician-diagnosed disease (top 3): Muscular and skeletal diseases: 555 diagnoses per 100 insured years Mental disorders: 549 diagnoses per 100 insured years Respiratory diseases: 324 diagnoses per 100 insured years |
| Otto et al. (2019) [35] | 142 (Subgroup of a cross-setting study) | 1. M±SD = 40.70±12.22 | Physical health: score incuding physical functioning, role-physical, bodily pain and general health | Physical health: physical health score [min: 0; max: 100]: M±SD = 48.23±9.80 |
| | | 2. Not reported by the authors | Mental health: score including vitality, social functioning, role-emotional, mental health; Stress | Mental health: Mental health score [min: 0; max: 100]: M±SD = 46.36±10.33 Stress [min: 0; max: 48]: M±SD = 17.82±10.64 |
| Rothgang et al. (2020) [52] | 674 (subgroup of nurses in LTC facilities in a study with several professions) | 1. not reported for the subsample nurses in LTC facilities 2. 80.7% (nurses working in LTC); | Physical health: complaints during/after work, physician-diagnosed disease | Complaints during/after work (top 3): Prevalence of low back pain: 64% Prevalence of physical excaustion: 62% Prevalence of pain in arms/hands: 58% Physician-diagnosed disease (top 3): Muscular and skeletal diseases: 476 diagnoses per 100 insured years Endocrine, nutritional and metabolic diseases: 284 diagnoses per 100 insured years Mental disorders: 284 diagnoses per 100 insured years |
| Schmidt (2010) [56] | 242 | 1. M±SD = 41.53±8.7 2. 82.6% | Mental health: burnout (emotional exhaustion, depersonalization), psychosomatic complaints | Mental health: Emotional exhaustion [min: 1; max: 6]: M±SD = 3.78±0.97 Depersonalization [min: 1; max: 6]: M±SD = 2.92±1.09 Psychosomatic complaints [min: 0; max: 3]: M±SD = 2.17±0.87 |
| Schmidt & Diestel (2011) [57] | 379 | 1. M±SD = 39.25±9.26 2. not reported | Mental health: burnout (emotional exhaustion), psychosomatic complaints | Mental health: Emotional exhaustion [min: 1; max: 6]: M±SD = 2.81±0.96 Psychosomatic complaints [min: 0; max: 3]: M±SD = 0.95±0.55 |

(*Continued*)

**Table 3.** (Continued)

| Long-term care (LTC) facilities | | | | |
|---|---|---|---|---|
| Author (year) | Sample | | Health problem, violence experience and related and outcome | Result |
| | Sample size (respondents) | 1. Age [years] 2. Gender (female) | | |
| Vaupel et al. (2021) [44] | 292 | not reported for the subgroup of nurses in LTC facilities | Violence: verbal and nonverbal sexual harassment and violence | Violence: Prevalence of nonverbal sexual harassment (at least one tima/year): 63.0% Prevalence of verbal sexual harassment and violence (at least one tima/year): 69.0% Prevalence of physical sexual harassment and violence (at least one tima/year): 53.0% |
| | | | Mental health: Burnout (emotional exhaustion), depressiveness, psychosomatic complaints, well-being (WHO 5) | Mental health: Prevalence of emotional exhaustion (at least one time/month): 58.4% Prevalence of depressiveness (often/ very often): 2.1% Prevalence of psychosomatic complaints (every few months to daily): 94.4% Prevalence of well-being (never to rarely): 13.0% |
| Wirth et al. (2017) [39] | 274 (Subgroup of a cross-setting study) | 1. M±SD = 44±11.8 | Mental health: score (Burnout, cognitive stress symptoms) | Mental health: Burnout [min: 0; max: 100]: M ±SD = 55±21 Cognitive Stress symptoms [min: 0; max: 100]: M±SD = 38±21 |
| | | 2. 83.1% | Physical health: health status | Physical health: Health status [min: 0; max: 100]: M ±SD = 63±20 |
| | | | Violence: physical, verbal | Violence: Physical Violence: 69% Verbal violence: 80.8% |
| Wollesen et al. (2019) [58] | 195 | 1. M±SD = 40.1±12.2 | Physical health: physical well-being | Physical health: Physical well-being [min: 0; max: 100]: M±SD = 43.38±8.68 |
| | | 2. 85.64% | Mental health: psychological well-being, stress level | Mental health: Psychological well-being [min: 0; max: 100]: M±SD = 45.92±10.81 Stress [min: 0; max: 48]: M ±SD = 18.76 ±10.36 |

contexts of the settings. In addition to common characteristics in the working conditions of the settings (e.g., time pressure, the availability or appropriate use of ergonomic equipment) further occupational exposure factors emerge. The daily work routine in LTC facilities and hospitals is typically characterized by work interruptions what, for example, points out the need for a coordinated cooperation with other professional groups. In contrast, nurses working in home-based LTC are very much on their own in terms of performing nursing activities and in some cases are also responsible for planning the work tours [59–61]. Although this might appear to be an advantage at first glance, this can easily become a stress factor, for example, if, despite legal requirements, no adequate breaks are taken or possible due to the necessary travel times and possible traffic problems [59–61]. In contrast, hospitals often have a strong hierarchical organizational structure resulting in comparably low participation opportunities for nurses [59, 60, 62]. The consideration of these different working conditions

**Table 4. Home-based long-term care—Summary of the studies included in the review.**

| Home-based long-term care | | | | |
|---|---|---|---|---|
| Author (year) | Sample | | Health problem, violence experience and related and outcome | Results |
| | Sample size | 1. Age [years] | | |
| | (respondents) | 2. Gender (female) | | |
| Gencer et al. (2019) [33] | 56 (Subgroup of a cross-setting study) | not reported for the subgroup of nurses in home-based long-term care | Violence: patient aggression | Violence<br>Patient aggression (6-fold likert scale [min: 0; max: 5]: 0: no strain; 5: high strain): M = 1.9 |
| | | | Physical health: subjective general health status | Physical health<br>No results for subgroup reported |
| | | | Mental health: score | Mental health<br>No results for subgroup reported |
| Otto et al. (2019) [35] | 20 (Subgroup of a cross-setting study) | 1. M±SD = 30.20±11.17 | Physical health: score incuding physical functioning, role-physical, bodily pain and general health | Physical health:<br>Physical health score [min: 0; max: 100]: M±SD = 54.77±5.76 |
| | | 2. not reported | Mental health: score including vitality, social functioning, role-emotional, mental health; Stress | Mental health:<br>Mental health score [min: 0; max: 100]: M±SD = 44.40±12.21<br>Stress: [min: 0; max: 48]: M±SD = 26.10±12.86 |
| Wirth et al. (2017) [39] | 92 (Subgroup of a cross-setting study) | 1. M±SD = 45.7±11.4 | Mental health: score (Burnout, cognitive stress symptoms) | Mental health:<br>Burnout [min: 0; max: 100]: M±SD = 46±24<br>Cognitive Stress symptoms [min: 0; max: 100]: M±SD = 27±22 |
| | | 2. 90.1% | Physical health: health status | Physical health:<br>Health status [min: 0; max: 100]: M±SD = 66±19 |
| | | | Violence: physical, verbal | Violence:<br>Physical Violence: 20.7%<br>Verbal violence: 70.3% |
| Vaupel et al. (2021) [44] | 107 | not reported for the subgroup of nurses in home-based long-term care | Violence: verbal and nonverbal sexual harassment and violence | Violence:<br>Prevalence of nonverbal sexual harassment (at least one tima/year): 48.1%<br>Prevalence of verbal sexual harassment and violence (at least one tima/year): 71.0%<br>Prevalence of physical sexual harassment and violence (at least one tima/year): 51.0% |
| | | | Mental health: Burnout (emotional exhaustion), depressiveness, psychosomatic complaints, well-being (WHO 5) | Mental health:<br>Prevalence of emotional exhaustion (at least one time/month): 50.0%<br>Prevalence of depressiveness (often/very often): 1,9%. Prevalence of psychosomatic complaints (every few months to daily): 98.1%<br>Prevalence of well-being (never to rarely): 7.5% |

provides one reason for the necessity of setting-specific differentiation in research on the health status and health behavior of nurses. On the other hand, this perspective offers starting points for putting the above-mentioned results of our review into context.

Although the differences in health problems appear to be small on the basis of the current data or have not yet been sufficiently systematically analyzed, it is widely acknowledged that the nursing profession differs significantly from other professions in terms of mental and physical health problems, as well as days of sick leave [36, 50, 52, 59]. In this respect, the available studies of our review are in line with current research showing high prevalences of burnout risk [39, 44, 46, 51, 53, 63] and musculoskeletal disorders [34, 40, 42, 52], independently from the setting nurses work in. This indicates that nurses do not only show an increased risk of long-term absence from work, but also are at risk in terms of occupational disability [13, 58, 64] and early retirement [8]. Thus, it is not surprising that in Germany almost every second employee in this field thinks about leaving the profession several times a year [9].

As a basis for a systematic analysis of the health problems of nurses in Germany, most data are available in the field of mental health, with most of the data on burnout, stress and psychosomatic complaints. Despite different operationalizations, the included studies reveal moderate to high burnout levels or a high burnout prevalence, respectively [39, 44, 46, 51, 53]. This is usually explained by the circumstances of the nursing profession, e.g., unfavorable working hours, routinely coping with obligatory rotating shifts, work overload because of understaffing, time pressure, interfacing problems with other occupational groups and high social responsibility [36, 59, 65–67]. Our results therefore are in line with current research and political demands on developing interventions or therapies helping to prevent or attenuate the above symptoms. Nevertheless, more setting-specific data are needed to help nurses manage their job-related tasks in a health-conscious manner. This also relates to the high proportion of musculoskeletal diseases. Our results show that especially musculoskeletal problems in the shoulder-neck area and low back pain take a high priority [34]. In principle, these problems are attributed to frequent and heavy lifting, as well as patient transfer and positioning [36, 59]. However, due to the different working conditions, setting-specific data would also be highly relevant in this regard for improving the working situation of nursing employees.

In our review, the topic of violence and the lack of studies on it were particularly striking. Compared to mental and physical health topics, violence was by far the least investigated. In total, only seven studies addressed violence experienced by nurses. Setting-specific information on this can only be derived from publications each. Going in line with other research [22], the available data indicate a high level of problems related to verbal and physical experiences of violence, as well as sexual harassment. The high prevalence of experiences of violence is also confirmed by international findings on this topic in acute medical care hospitals [68]. For a well-founded setting-specific comparison, however, the data available in Germany is still very scarce and international comparisons are very limited due to the different health care systems. Therefore, a significant research gap in order to contribute to an improvement in working conditions for nurses is considered. Assuming that violence experience represents a high psychological burden for nurses, the topic of violence prevention and dealing with experiences of violence must be attributed a central role in workplace health promotion measures. At present, it can be assumed that a systematic reappraisal rarely takes place in this regard [22].

Due to the high health burdens and the well-known social and political relevance of the nursing profession, which since the pandemic has been counted among the so-called "system-relevant professions", the *Nursing Personnel Strengthening Act* [69], as well as the *Concerted Action on Nursing* [70] intend to support an improvement in working conditions in nursing. The aim is, amongst other things, to promote support options for professional nursing staff who are exposed to physical or mental stress and to increase the attractiveness of the nursing profession. This, however, requires a systematic and setting-related analysis of the health situation of professional nurses. In this respect, our results indicate that the field of home-based LTC is by far the least studied compared to LTC facilities for the elderly and acute hospitals.

## Strengths and limitations of the review

The results of our review provide a setting-specific and cross-setting insight into the current evidence about health problems and violence experiences of nurses in Germany. Thereby, it highlights knowledge gaps that need to be addressed to improve the setting-specific working conditions for nurses. Nevertheless, some limitations occur. The conclusions of this review are limited due to the lack of comparable setting-specific data. Thereby, it needs to be considered that the number of studies with setting-specific data differs substantially. While most studies are available for the setting "acute care hospital", there are hardly any for the setting

"home-based LTC". In addition, our research was limited to adult care and our definition of settings had to be specified (e.g., exclusion of community-based LTC settings). Current evidence about health problems of nurses working in those settings should be addressed in future studies. Furthermore, the results obtained are subject to variability in study design and/or measurement instruments used. Additionally, the very different and in some cases extremely small sample sizes or subsample sizes of the different studies also significantly limited the comparability of the included studies. Due to the limited data available, no further specification about nurses (e.g., according to nursing degrees), different care areas within the setting (e.g., palliative care units, intensive care units), or patients (e.g., level of care, musculoskeletal disorders) was possible. Beyond, our research might be affected by a publication bias, since it was limited to scientific publications and health insurance reports which were considered as grey literature. Occupational and health science projects that could not be found in one of the scientific databases were not taken into account what indicates a risk of publication bias. Furthermore, misinterpretation by authors of available data cannot be completely avoided [71].

## Conclusion

This review provides an overview of the current state of research on setting-specific data on health problems and violence experiences of nurses in Germany. Considering the socio-economic and political relevance of this profession, we argue that it is crucial to get insight in setting-specific differences of nurses' health problems and violence experiences. The aim of our review was to examine regular health issues affecting nurses in Germany, including physical and mental health. Since violence experiences are apparent in nurses' everyday working life and are strongly related to physical and mental health, the previously neglected topic of violence experiences was also explicitly taken up.Hence, the state of research on this topic is characterized by a lack of studies explicitly comparing the three settings. Beyond, it is characterized by heterogeneity of health problems assessed, operationalization and sample size. This makes it difficult to compare studies within a setting and across settings. Due to the high relevance in practice, the clear underrepresentation of data and studies on the topic of violence experiences of nurses is also worth considering. We hope that our review will help to underline the need for target-group specific occupational health interventions for nurses in different settings. Furthermore, we emphasize the importance of a sound empirical basis for this, taking into account setting-specific aspects and violence experiences. On this basis, occupational health interventions could be developed or it could be examined whether interventions applied in practice adequately address the needs of nurses.

## Supporting information

**S1 Checklist. PRISMA checklist.**
(DOC)

**S1 File. Quality assessment.**
(DOCX)

**S2 File. Search strategy.**
(DOCX)

## Author Contributions

**Conceptualization:** Andrea Schaller.

**Data curation:** Teresa Klas, Madeleine Gernert.

**Formal analysis:** Teresa Klas, Madeleine Gernert.

**Funding acquisition:** Andrea Schaller.

**Methodology:** Andrea Schaller, Teresa Klas, Madeleine Gernert.

**Supervision:** Andrea Schaller.

**Visualization:** Andrea Schaller, Teresa Klas, Madeleine Gernert.

**Writing – original draft:** Andrea Schaller.

**Writing – review & editing:** Andrea Schaller, Teresa Klas, Madeleine Gernert, Kathrin Steinbeißer.

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
