## [Decision Letter · Decision Letter 0]

20 Sep 2021

PONE-D-21-22976Health problems and violence experiences of nurses working in acute care hospitals, long-term care facilities or home-based long-term care: a systematic review.PLOS ONE

Dear Dr. Schaller,

Thank you for submitting your manuscript to PLOS ONE. After careful consideration, we feel that it has merit but does not fully meet PLOS ONE’s publication criteria as it currently stands. Therefore, we invite you to submit a revised version of the manuscript that addresses the points raised during the review process.

We look forward to receiving your revised manuscript.

Kind regards,

Jenny Wilkinson, PhD

Academic Editor

PLOS ONE

Journal Requirements:

2. Please confirm that you have included all items recommended in the PRISMA checklist including:

- the full electronic search strategy used to identify studies with all search terms and limits for at least one database.

- an explanation for why the search inclusion dates began in 2009

- a Supplemental file of the results of the individual components of the quality assessment, not just the overall score, for each study included.

- See https://journals.plos.org/plosmedicine/article?id=10.1371/journal.pmed.1000100#pmed-1000100-t003 for guidance on reporting.

Thank you.

3. We note that you have stated that you will provide repository information for your data at acceptance. Should your manuscript be accepted for publication, we will hold it until you provide the relevant accession numbers or DOIs necessary to access your data. If you wish to make changes to your Data Availability statement, please describe these changes in your cover letter and we will update your Data Availability statement to reflect the information you provide

Reviewers' comments:

Reviewer's Responses to Questions

**Comments to the Author**

1. Is the manuscript technically sound, and do the data support the conclusions?

Reviewer #1: Yes

Reviewer #2: No

2. Has the statistical analysis been performed appropriately and rigorously? 

Reviewer #1: N/A

Reviewer #2: No

3. Have the authors made all data underlying the findings in their manuscript fully available?

Reviewer #1: Yes

Reviewer #2: Yes

4. Is the manuscript presented in an intelligible fashion and written in standard English?

Reviewer #1: Yes

Reviewer #2: No

5. Review Comments to the Author

Reviewer #1: Page 3, line 60: I suggest you specify that these data refer to the German nation.

Page 3, lines 73-79: it is very important to expand the literature supporting these statements to an international context and not just a German one, the only non-German article cited is dated (2000). Some studies are suggested below, in a non-exhaustive way:

Jiali Liu, Jing Zheng, Ke Liu, Xu Liu, Yan Wu, Jun Wang, Liming You. Workplace violence against nurses, job satisfaction, burnout, and patient safety in Chinese hospitals. Nurs Outlook. 2019;67(5):558-566. doi: 10.1016/j.outlook.2019.04.006;

Shi-Hong Zhao, Yu Shi, Zhi-Nan Sun, Feng-Zhe Xie, Jing-Hui Wang, Shu-E Zhang, Tian-Yu Gou, Xuan-Ye Han, Tao Sun, Li-Hua Fan. Impact of workplace violence against nurses' thriving at work, job satisfaction and turnover intention: A cross-sectional study. J Clin Nurs. 2018;27(13-14):2620-2632. doi: 10.1111/jocn.14311.

Page 4, lines 94-95: it is advisable to add a supporting bibliography, for example:

Ferri Paola, Stifani Serena, Accoto Angela, Bonetti Loris, Rubbi Ivan, Di Lorenzo Rosaria. Violence Against Nurses in the Triage Area: A Mixed-Methods Study. J Emerg Nurs. 2020;46(3):384-397. doi: 10.1016/j.jen.2020.02.013;

Li Lu, Ka-In Lok, Ling Zhang, Ailing Hu, Gabor S Ungvari, Daniel T Bressington, Teris Cheung, Feng-Rong An, Yu-Tao Xiang. Prevalence of verbal and physical workplace violence against nurses in psychiatric hospitals in China. Arch Psychiatr Nurs. 2019;33(5):68-72. doi: 10.1016/j.apnu.2019.07.002.

Page 4: I would suggest to clarify the rationale of the systematic review, in particular because it was chosen to compare different care settings and limit it only to the German context.

Page 20, lines 229-231 “With regard to experiences of violence, nurses working in LTC facilities appear to be more frequently affected compared to those working in acute hospitals and home-based LTC”: It would be really important to compare these results to international studies.

Page 20, lines 252-254: “In this respect, the available studies of our review are in line with current research showing high prevalences of burnout risk” It is suggested to broaden the comparison with international studies that have investigated burnout in nurses, such as, for example, not exhaustive:

Chiara Dall'Ora, Jane Ball Maria Reinius, Peter Griffiths. Burnout in nursing: a theoretical review. Hum Resour Health. 2020;18(1):41. doi: 10.1186/s12960-020-00469-9.

Ferri P, Guerra E, Marcheselli L, Cunico L, Di Lorenzo R. Empathy and burnout: an analytic cross-sectional study among nurses and nursing students. Acta Biomed. 2015;86 Suppl 2:104-15.

Page 21, lines 257-258: “It is therefore not surprising that almost every second employee in this field thinks about leaving the profession several times a year [9]” :

A Nantsupawat, W Kunaviktikul, R Nantsupawat, O-A Wichaikhum, H Thienthong, L Poghosyan. Effects of nurse work environment on job dissatisfaction, burnout, intention to leave. Int Nurs Rev. 2017;64(1):91-98. doi: 10.1111/inr.12342.

Page 25, line 391 “Rippon TJ. Aggression and violence in health care professions. J Adv Nurs. 2000; 31:452–

392 60. doi: 10.1046/j.1365-2648.2000.01284.x PMID: 10672105” it is not necessary, in addition to the doi, to report the PMID.

Finally, a revision of the English translation is suggested.

Reviewer #2: Please note the attachment. This Systematic review requires major revision as I enclosed in the attached document. Please revise the methods section and follow the recommended checklist to ensure that a high quality of your study achieved.

6. PLOS authors have the option to publish the peer review history of their article (what does this mean?). If published, this will include your full peer review and any attached files.

Reviewer #1: No

Reviewer #2: No

---

## [Author Response · Author response to Decision Letter 0]

15 Oct 2021

Thank you very much for pointing out the potential of our manuscript and the opportunity to revise it based on your comments. Your comments helped us a lot to substantially improve the quality of our manuscript. Please find below our point-by-point answers. Marked passages highlight the changes in the revised manuscript. 

Editor comments:

• Thanks for the advice. We have taken the guidelines into account

2. Please confirm that you have included all items recommended in the PRISMA checklist including:

- the full electronic search strategy used to identify studies with all search terms and limits for at least one database.

- an explanation for why the search inclusion dates began in 2009

- a Supplemental file of the results of the individual components of the quality assessment, not just the overall score, for each study included.

• The full electronic search strategy for the PubMed database is submitted in a supplemental file with the revision

• Please apologize our mistake in the abstract. As stated in the methods section of the original version, “Original studies in German or English language published between January 01st, 2010 and January 11th, 2021 were taken into account (p. 4, lines 117 – 118). We have narrowed our search to the last 10 years, as an even longer period would not be adequate regarding the health policy framework conditions in Germany. we have corrected the year in the abstract: 

o “Articles were included if they had been published after 2010 and provided data on health problems or violence experiences of nurses in at least one care setting.” (p. 2, lines 32-34)

• We have enclosed a supplemental file containing the quality assessment for each study included.

3. We note that you have stated that you will provide repository information for your data at acceptance. Should your manuscript be accepted for publication, we will hold it until you provide the relevant accession numbers or DOIs necessary to access your data. If you wish to make changes to your Data Availability statement, please describe these changes in your cover letter and we will update your Data Availability statement to reflect the information you provide

• Please excuse the incorrect information. All relevant data are within the paper and its Supporting Information files.

 

Reviewers' comments:

1. Is the manuscript technically sound, and do the data support the conclusions?

Reviewer #1: Yes; Reviewer #2: No

• We have revised the manuscript according to the feedback from Reviewer 2 (see below in the detailed comments)

2. Has the statistical analysis been performed appropriately and rigorously? 

Reviewer #1: N/A; Reviewer #2: No

• We have revised the manuscript according to the feedback from Reviewer 2 (see below in the detailed comments)

3. Have the authors made all data underlying the findings in their manuscript fully available?

Reviewer #1: Yes; Reviewer #2: Yes

4. Is the manuscript presented in an intelligible fashion and written in standard English?

Reviewer #1: Yes; Reviewer #2: No

• The manuscript was again revised regarding language. Changed parts are highlighted within the manuscript.

5. Review Comments to the Author

 

Reviewer #1: 

#1.1: Page 3, line 60: I suggest you specify that these data refer to the German nation.

• We added the following information: 

“In 2019, more than 19 million medical cases were treated in German hospitals, including acute medical care hospitals and rehabilitation facilities [2].” (p. 3, lines 61-62)

#1.2: Page 3, lines 73-79: it is very important to expand the literature supporting these statements to an international context and not just a German one, the only non-German article cited is dated (2000). Some studies are suggested below, in a non-exhaustive way:

• Many thanks for this advice and the references. We have expanded the section as follows: 

“Violence experiences can include physical or verbal violence experiences, patient- or relatives related aggressions and sexual harassment [15,16]. Violence experiences can lead not only to physical harm but also to mental health problems including impaired well-being or even symptoms of post-traumatic stress disorder [17–19]. Additionally, nurses’ feelings of anxiety and anger due to violence experiences also go in line with less job satisfaction [20] and enhanced withdrawal intentions [21].” (p. 3, lines 77-83)

• Newly considered international literature sources: 

o 17. Franz S, Zeh A, Schablon A, Kuhnert S, Nienhaus A. Aggression and violence against health care workers in Germany--a cross sectional retrospective survey. BMC Health Serv Res. 2010; 10:51. Epub 2010/02/25. doi: 10.1186/1472-6963-10-51 PMID: 20184718.

o 18. Willness CR, Steel P, Lee K. A Meta-Analysis of the Antecedents and Consequences of workplace sexual harassment. Personnel Psychology. 2007; 60:127–62. doi: 10.1111/j.1744-6570.2007.00067.x.

o 19. Bowling NA, Beehr TA. Workplace harassment from the victim's perspective: a theoretical model and meta-analysis. J Appl Psychol. 2006; 91:998–1012. doi: 10.1037/0021-9010.91.5.998. PMID: 16953764.

o 20. Liu J, Zheng J, Liu K, Liu X, Wu Y, Wang J, et al. Workplace violence against nurses, job satisfaction, burnout, and patient safety in Chinese hospitals. Nurs Outlook. 2019; 67:558–66. Epub 2019/05/02. doi: 10.1016/j.outlook.2019.04.006 PMID: 31202444.

o 21. Barling J, Rogers AG, Kelloway EK. Behind closed doors: in-home workers' experience of sexual harassment and workplace violence. J Occup Health Psychol. 2001; 6:255–69.

#1.3: Page 4, lines 94-95: it is advisable to add a supporting bibliography, for example:

o We added the following references: to the sentence:

“As nurses’ daily working life differs over the settings, it is assumed that health problems and violence experiences might be different, too [25,26].” (p. 4, line 100)

25. Galdikien N, Asikainen P, Balčiūnas S, Suominen T. Do nurses feel stressed? A perspective from primary health care. Nursing & Health Sciences. 2014; 16:327–34. doi: 10.1111/nhs.12108 PMID: 25389543.

26. Ferri P, Silvestri M, Artoni C, Di Lorenzo R. Workplace violence in different settings and among various health professionals in an Italian general hospital: a cross-sectional study. Psychol Res Behav Manag. 2016; 9:263–75. Epub 2016/09/23. doi: 10.2147/PRBM.S114870 PMID: 27729818.

#1.4: Page 4: I would suggest to clarify the rationale of the systematic review, in particular because it was chosen to compare different care settings and limit it only to the German context.

• We hope to bring out the rationale or background more clearly with the following additions and additional references in the background:

o “Despite the international differences in health care systems, professional nursing is generally described as a crucial part of the health care system. It encompasses health promotion, disease prevention, and care of individuals of all ages with physical or mental illness, or with disabilities [1]. In the German care system, nursing care takes place in different settings, such as in hospitals, nursing homes, at home or in community-based institutions. Crucial settings for adult-based nursing care are acute care hospitals, long-term care facilities (LTC) and the patients’ home.” (p. 3, lines 54 – 60)

o “Violence experiences can lead not only to physical harm but also to mental health problems including impaired well-being or even symptoms of post-traumatic stress disorder [17–19].” (p.3, lines 79 – 83).

o “As nurses’ daily working life differs over the settings, it is assumed that health problems and violence experiences might be different, too [25,26].” (p. 4, lines 99 – 100)

#1.5: Page 20, lines 229-231 “With regard to experiences of violence, nurses working in LTC facilities appear to be more frequently affected compared to those working in acute hospitals and home-based LTC”: It would be really important to compare these results to international studies.

• Thank you for the comment. We have added the following international review, in order to place the results in the international context:

o 68. Liu J, Gan Y, Jiang H, Li L, Dwyer R, Lu K, et al. Prevalence of workplace violence against healthcare workers: a systematic review and meta-analysis. Occup Environ Med. 2019; 76:927–37. Epub 2019/10/13. doi: 10.1136/oemed-2019-105849 PMID: 31611310.

o “Going in line with other research [22], the available data indicate a high level of problems related to verbal and physical experiences of violence, as well as sexual harassment. The high prevalence of experiences of violence is also confirmed by international findings on this topic in acute medical care hospitals [68]. For a well-founded setting-specific comparison, however, the data available in Germany is still very scarce and international comparisons are very limited due to the different health care systems. Therefore, a significant research gap in order to contribute to an improvement in working conditions for nurses is considered.” (p. 21, lines 292 – 299)

#1.6: Page 20, lines 252-254: “In this respect, the available studies of our review are in line with current research showing high prevalences of burnout risk” It is suggested to broaden the comparison with international studies that have investigated burnout in nurses, such as, for example, not exhaustive:

• We added this reference in the discussion:

o “In this respect, the available studies of our review are in line with current research showing high prevalences of burnout risk [39,44,46,51,53,63] and musculoskeletal disorders [34,40,42,52], independently from the setting nurses work in.” (p. 20, lines 266 – 269)

o 63 Dall'Ora D, Jane Ball Maria Reinius, Peter Griffiths. Burnout in nursing: a theoretical review. Hum Resour Health. 2020;18(1):41. doi: 10.1186/s12960-020-00469-9. 

#1.7: Page 21, lines 257-258: “It is therefore not surprising that almost every second employee in this field thinks about leaving the profession several times a year [9]”:

• Since our results are limited to the German health care system, we prefer to limit ourselves to a German reference for this statement because of the international differences in health care systems. We have highlighted this as follows:

“Thus, it is not surprising that in Germany almost every second employee in this field thinks about leaving the profession several times a year [9].” (p. 21, lines 271–272)

#1.8: Page 25, line 391: it is not necessary, in addition to the doi, to report the PMID.

• Thank you for pointing this out. We have followed the journal guidelines and will consult with the editorial team if the article is to be published.

#1.9: Finally, a revision of the English translation is suggested.

• We had a colleague revise the language of the manuscript.

 

Reviewer #2:

2.1: I almost felt that these was two systematic reviews in one. It could be “Health problems of nurses” and “Violence Experiences among nurses” I am not sure what is the relation between both? Why to include both under one systematic review? 

• Thank you very much for this important comment. We hope to bring this out more clearly with the following additions and additional references in the background:

o “Violence experiences can lead not only to physical harm, but also to mental health problems including impaired well-being or even symptoms of post-traumatic stress disorder [17–19]. Additionally, nurses’ feelings of anxiety and anger due to violence experiences also go in line with less job satisfaction [20] and enhanced withdrawal intentions [21]..” (p.3, 79 – 83).

o “As nurses’ daily working life differs over the settings, it is assumed that health problems and violence experiences might be different, too [25,26].” (p. 4, 99 – 100)

• Beyond, we expanded the conclusion as follows: “

o “The aim of our review was to examine regular health issues affecting nurses in Germany, including physical and mental health. Since violence experiences are apparent in nurses’ everyday working life and are strongly related to physical and mental health, the previously neglected topic of violence experiences was also explicitly taken up.” (p.23, lines 342 – 346)

#2.2: Title: the word “or” should be “and” as both settings were reviewed by the authors. Also add “in Germany” under the title as the authors excluded studies outside Germany. 

• We have revised the title as follows:

Health problems and violence experiences of nurses working in acute care hospitals, long-term care facilities, and home-based long-term care in Germany: A systematic review.

#2.3: Introduction: should be rewritten and it will benefit from details explanation on the impact of workplace violence and its risk factors and why to include this topic under systematic review? What is the gap in the current literature in this regard? What is the aim of this systematic review?

• As explained under comment 2.1, we hope to bring this out more clearly with the following additions and additional references in the background:

o “Violence experiences can lead not only to physical harm but also to mental health problems including impaired well-being or even symptoms of post-traumatic stress disorder [17–19].” (p.3, lines 79 – 83).

o “As nurses’ daily working life differs over the settings, it is assumed that health problems and violence experiences might be different, too [25,26].” (p. 4, lines 99 – 100)

#2.4: Under introduction Line 60: the statement “In 2018, 19.4 million patients were treated in hospitals” Which hospitals?

• Thank you for this advice. We have revised the sentence as follows:

“In 2019, more than 19 million medical cases were treated in German acute medical care hospitals and rehabilitation facilities [2].” (p. 3, lines 61-62)

#2.5: Under introduction Line 77: “Theycan” should read “They can” 

• Thank you very much, we have followed the suggestion by rewording the content as follows: 

“Violence experiences can include physical or verbal violence experiences, patient- or relatives related aggressions, and sexual harassment [15,16].” (p. 3, lines 77 – 79)

Methods: 

#2.6: Search strategy: Why only used PubMed and PubPsych? How about other database such as Medline, CINAHL, Web of sciences database? 

• Since the PRISMA guidelines do not specify the number of databases, we agreed on two different databases. In addition, MEDLINE was integrated into PUBMed search. We excluded CINAHL as a fee-based database.

#2.7: Explain in details Keywords and MESH. What are the Boolean operators used in this study? 

• We pointed out the keywords and operators more clearly in the text: 

“Search terms used for relevant studies were built of the following keywords and Boolean operators (in cursive): (nurs* OR "professional care" OR "professional caregiver") AND (health OR violence) AND (“cross sectional” OR survey) AND (german*).” (p. 4, lines 115 – 118)

• We did not consider MeSH. After reviewing the terms listed in MeSH, this would have inappropriately extended our very specific search strategy without any gain in relevant studies.

#2.8: What are the participants used in the search? Did they include nurse practitioners, supervisors, and managers of nursing, ….etc?

• Our participants were professional nurses in Germany. Apprentices, supervisors and managers were excluded. We pointed this out more clearly:

• “Studies which met at least one of the following criteria were excluded: (1) longitudinal studies or validation studies, (2) qualitative studies, (3) studies outside of Germany. Additionally, we excluded studies addressing health issues of apprentices, supervisors, or managers” (p.5, lines 130 – 131) 

#2.9: Authors should not used health insurance reports as these unreliable from scientific point of view. Only published studies can be used in Systematic Reviews.

• We have followed the PRISMA, which do not prescribe any quality standards with regard to study selection. Thereby, the consideration of grey literature, such as reports, is considered desirable. We added this aspect in the limitations:

“Beyond, our research might be affected by a publication bias, since it was limited to scientific publications and health insurance reports which were considered as grey literature.” (p. 22, lines 332 – 334)

#2.10: Inclusion and Exclusion criteria: should be explained in detail. Why to include only cross-sectional studies? These subjected to bias. What are skilled nurses in Germany? Why they excluded longitudinal studies and studies outside of Germany? 

• The aim of our review was to get an overview about the current health problems and violence experiences of nurses in different care settings in Germany. To answer this question, we consider the focus on cross-sectional data to be justifiable. We would like to remark at this point that we did not intend to conduct a Cochrane Review of intervention studies.

• Studies outside Germany were excluded due to international differences in the health care systems.

• Please excuse the ambiguity: by skilled nurses we understand professional nurses, qualified by graduation from an accredited school of nursing and by passage of a national licensing examination to practice nursing. We have replaced the term “skilled nurses” by “professional nurses” in the manuscript and added the previous definition in the methods section: 

o “In the present study, professional nurses were considered to be qualified by graduation from an accredited school of nursing and by passage of a national licensing examination to practice nursing.” (p. 5, lines 155-157)

#2.11: Quality assessment: JBI checklist of prevalence studies is not appropriate to use in this review. The two independent reviewers should assess the included studies using the Critical Appraisal Skills Program (CASP) checklist (Critical Appraisal Skills Program (CASP) Checklist, 2016) in two phases. During the first phase, each reviewer read the title and abstract of all the citations retrieved and entered this information into a custom-designed database. In the second phase, full-text articles that met the inclusion criteria should be retrieved and reviewed. Pertinent information related to health problems and violence experiences among nurses should be collected and stored in the database. The CASP used to evaluate the methodological rigor of each of the studies. Also use the Cochrane Collaboration ‘Risk of bias’ assessment tool across six domains of bias including selection, performance, detection, attrition, presorting and other (Higgins, Altman, & Sterne, 2011). Furthermore, the inter-rater reliability should be assessed by Kappa statistical test. Then authors should report the Kappa level? 

• Thank you for this remark. We decided to keep the JBI checktlist, which we would like to explain below:

o Again, we would like to remark at this point that we did not intend to conduct a Cochrane Review of intervention studies. The Critical Appraisal Skills Program (CASP) checklists are not applicable to the included study designs. As we have only considered cross-sectional data to represent the prevalence of health problems, we consider the JBI checklist for prevalence studies as appropriate. It is tailored to cross-sectional studies and was applied with a four-eyes-principle.

o We think that the same applies to the risk of bias assessment tool of the Cochrane Collaboration, which is designed for intervention studies. Five of the six bias dimensions mentioned are therefore not applicable to our primary studies (selection, performance, detection, attrition, and presorting bias). 

o We have addressed the sixth bias dimension (“other sources of bias”) in the limitations:

“Additionally, the very different and in some cases extremely small sample sizes or subsample sizes of the different studies also significantly limited the comparability of the included studies. Due to the limited data available, no further specification about nurses (e.g., according to nursing degrees), different care areas within the setting (e.g., palliative care units, intensive care units), or patients (e.g., level of care, musculoskeletal disorders) was possible. Beyond, our research might be affected by a publication bias, since it was limited to scientific publications and health insurance reports which were considered as grey literature. Occupational and health science projects that could not be found in one of the scientific databases were not taken into account what indicates a risk of publication bias. Furthermore, misinterpretation by authors of available data cannot be completely avoided [71].” (p. 22, lines 327-336)

#2.12: Study selection: What is the definition of violence included in this Systematic Review (SR)? Type of violence? Did the authors explored whether the reporting of 29 studies associated with being Cochrane review?

• Thank you for this remark. We included the types of violence in the inclusion criteria and added a definition of violence:

o (4) data on physical health, mental health, and/or violence experience (physical or verbal violence experiences, patient-related aggressions, sexual harassment) (p 5, lines 126-1289).

o “Our definition of violence is based on the WHO definition of violence against patients or residents. This definition is acknowledged and accepted in the field of nursing and comprises emotional, physical and sexual forms of violence which cause harm or distress to the affected person [30,31]..” (p. 5 – 6, lines 157-160)

Data extraction

#2.13: Authors should perform data extraction on a random sample of the included systematic Review (SR) which were selected using the random number generator. Sampling should be stratified so that the proportion of Cochrane reviews in the selected sample equaled that in the total sample

To minimize errors in the remaining sample of SR, one author verifies the data for these items in all SR. Also, one author reviewed the free text responses of all items with an “Other (please specify)” option. Responses should be modified if it was judged that one of the forced-choice options was a more appropriate selection.

• As we did not conduct a Cochrane Review (see 2.10 and 2.11) we have followed the Prisma guidelines for data extraction. This procedure was reported as follows:

“Extracted data included the setting in which the study was conducted (acute care hospital, LTC facilities, home-based LTC, cross-setting), author and publication year, sample size, sample characteristics (age, gender), the health problem and/or violence experience assessed in the study (physical health, mental health, and/or violence experiences) and the findings of the study related to the respective health problem. In the present study, professional nurses were considered to be qualified by graduation from an accredited school of nursing and by passage of a national licensing examination to practice nursing. Our definition of violence is based on the WHO definition of violence against patients or residents. This definition is acknowledged and accepted in the field of nursing and comprises emotional, physical and sexual forms of violence which cause harm or distress to the affected person [30,31]. Some of the included studies did not contain all the aforementioned variables. In these cases, the available data were reported. Missing data was indicated by the note "not reported". (p.5 - 6, lines 151-162):

Data synthesis

#2.14: Data synthesis should involve a mixture of descriptive summaries of the included methodological research papers. Data extracted from research articles that described a summary of measures (odds ratios, the difference in means, incident rate ratios) should be grouped and analyzed by study design. From this analysis, you should prepare a descriptive analysis of the included studies. Authors did not explain their data synthesis.

• Based on our research question and the different outcomes and measurement instruments in the primary studies, a synthesis of the quantitative data is not possible. As described in the methods section, we therefore intentionally limit ourselves to a narrative report of the results in addition to the tables:

“The tables were used as a basis for a narrative synthesis of the key findings of the included studies.” (p.6, lines 171 – 172)

Results:

#2.15: Figure 1 PRISMA Flow chart has many errors, and it is very confusing. For example, full text article in figure 1 were excluded with reasons 20….etc., these did not add up into n=37. Also 8 studies are missing to come down from 37 to 29. The total studies included under manual search were 22 and if these included in the final 29 we left with only 7 and how many of these either from Pubmed or PubPsych? Please clarify.

• We apologize for our mistake. The studies identified through manual search was 12. We corrected this number in the figure.

• In addition, we added the number of studies identified in each database:

“Of these, 15 studies were found in PubMed and 8 studies in PubPsych, whereby six duplicates occured.” (p.6., lines 179 – 180)

#2.16: Why to include studies with setting unspecific as reported under Table 1. These should be excluded from this SR as it contradicts with the aim of the study and the title of this SR. 

• Thank you for this advice. From our point of view, however, setting-unspecific or cross-setting studies do not contradict our research aim. In fact, more setting-specific studies would be desirable from our point of view, as this would allow an appropriate comparison of the health problems and violence experiences of professional nurses, provided that the subgroups are identified in detail and the results are reported accordingly.

#2.17: Tables should be revised as per the above comments 

• Since we are unfortunately unable to take most of the feedback on the presentation of results into account, we have decided not to revise the tables.

#2.18: Discussion: the statement “this is the first review” is very strong and authors unaware of unpublished work. I suggest deleting it

• Thanks for the advice. Please consider that we introduced our statement with “To our knowledge” (p. 20, line 235). In our opinion, this indicates that there may be unpublished works, for example, which we then cannot know about, of course.

#2.19: Limitations of the study: should explain in details and logical ways. Personal, misinterpretation, publication biases are possibilities in this SR. 

• Thank you for pointing this out. We have extended the limitations as follows: 

“Beyond, our research might be affected by a publication bias, since it was limited to scientific publications and health insurance reports which were considered as grey literature. Occupational and health science projects that could not be found in one of the scientific databases were not taken into account what indicates a risk of publication bias. Furthermore, misinterpretation by authors of available data cannot be completely completely avoided [71].” (p. 22, 332 – 336)

---

## [Decision Letter · Decision Letter 1]

2 Nov 2021

Health problems and violence experiences of nurses working in acute care hospitals, long-term care facilities, and home-based long-term care in Germany: A systematic review.

PONE-D-21-22976R1

Dear Dr. Schaller,

We’re pleased to inform you that your manuscript has been judged scientifically suitable for publication and will be formally accepted for publication once it meets all outstanding technical requirements.

Kind regards,

Sinan Kardeş, M.D.

Academic Editor

PLOS ONE

Reviewers' comments:

Reviewer's Responses to Questions

**Comments to the Author**

1. If the authors have adequately addressed your comments raised in a previous round of review and you feel that this manuscript is now acceptable for publication, you may indicate that here to bypass the “Comments to the Author” section, enter your conflict of interest statement in the “Confidential to Editor” section, and submit your "Accept" recommendation.

Reviewer #1: All comments have been addressed

Reviewer #2: All comments have been addressed

2. Is the manuscript technically sound, and do the data support the conclusions?

Reviewer #1: Yes

Reviewer #2: Yes

3. Has the statistical analysis been performed appropriately and rigorously? 

Reviewer #1: Yes

Reviewer #2: Yes

4. Have the authors made all data underlying the findings in their manuscript fully available?

Reviewer #1: Yes

Reviewer #2: Yes

5. Is the manuscript presented in an intelligible fashion and written in standard English?

Reviewer #1: Yes

Reviewer #2: Yes

6. Review Comments to the Author

Reviewer #1: The authors promptly revised the manuscript based on suggestions. The systematic review is very interesting and will contribute to knowledge on this important issue.

Reviewer #2: The required corrections have been made and the manuscript has been improved scientifically. The authors responded to my comment and provided valuable information which are important for the readers. I suggest accepting once the attached comments are incorporated in the manuscript

7. PLOS authors have the option to publish the peer review history of their article (what does this mean?). If published, this will include your full peer review and any attached files.

Reviewer #1: No

Reviewer #2: No

---

## [Editor Report · Acceptance letter]

8 Nov 2021

PONE-D-21-22976R1 

Health problems and violence experiences of nurses working in acute care hospitals, long-term care facilities, and home-based long-term care in Germany: A systematic review. 

Dear Dr. Schaller:

I'm pleased to inform you that your manuscript has been deemed suitable for publication in PLOS ONE. Congratulations! Your manuscript is now with our production department. 

Kind regards, 

on behalf of

Dr. Sinan Kardeş 

Academic Editor

PLOS ONE